# Pandemic Financial Stress in Dental Medicine in Croatia

**DOI:** 10.3390/dj11010009

**Published:** 2022-12-27

**Authors:** Edi Orlic, Stjepan Spalj, Natasa Ivancic Jokic, Danko Bakarcic, Odri Cicvaric, Renata Grzic

**Affiliations:** 1Private Dental Practice, 51000 Rijeka, Croatia; 2Department of Orthodontics, Faculty of Dental Medicine, University of Rijeka, 51000 Rijeka, Croatia; 3Department of Paediatric Dentistry, Faculty of Dental Medicine, University of Rijeka, 51000 Rijeka, Croatia; 4Department of Prosthodontics, Faculty of Dental Medicine, University of Rijeka, 51000 Rijeka, Croatia

**Keywords:** COVID-19, dentistry, financial stress, pandemics

## Abstract

The aim of this cross-sectional research was to investigate how the COVID-19 pandemic affected the activity of dental medicine in the Republic of Croatia in 2020. It included 136 doctors of dental medicine who completed an online survey regarding their personal and professional information; work in dental offices; and level of fear for their own health, the health of others, and financial existence; and their attitude about vaccination. There was a significantly higher decrease in patient visits in dental offices that do not have a contract with public health insurance (70% vs. 37%; *p* < 0.001) and in dental offices that have a higher percentage of profit from dental tourism (32% vs. 14%; *p* < 0.001). Fear of financial existence was significantly higher in the group of dentists who do not have a contract with public health insurance (*p* = 0.0) and is positively correlated with the percentage of profit from dental tourism (r = 0.299; *p* < 0.001). Dentists with a higher level of fear that they or their loved ones would get infected due to the nature of their job are more likely to get vaccinated (*p* ≤ 0.007). The decision to get vaccinated and wearing a disposable coat/apron was related to fear when all other parameters were controlled for (R = 0.44; *p* = 0.037). In conclusion, the COVID-19 pandemic had a minimal impact on the profession of dental medicine in Croatia but represented a larger financial stress for dentists working in dental offices that do not have a contract with public health insurance and have a higher percentage of income from dental tourism.

## 1. Introduction

COVID-19, an infectious disease caused by the SARS-CoV-2 virus, was discovered in China at the end of December 2019 [1]. On 11 March 2020, the World Health Organization declared the COVID-19 pandemic, which represents one of the largest public health problems in the world [2,3]. The SARS-CoV-2 virus has proven to be very easily transmissible among the human population around the world, so in the first two months of its appearance, more than 100,000 cases were reported. The main ways of transmission are by droplets, direct contact, and indirect contact via contaminated surfaces [4,5].

In the beginning, no one could assume that by 1 April 2022 in Croatia, almost 728,000 people would be infected with the SARS-CoV-2 virus and 12,699 people would have died. On a global scale, the figures are 300,000,000 infected and 5,500,000 dead. The first case of the disease in Croatia was detected at the end of February 2020, which was imported from the Italian province of Lombardy, which was affected by the epidemic [6]. Soon, the national crisis staff within administrative and health services started implementing preventive measures directed towards the contacts of the infected people in order to reduce the risk of spreading the infection. At the same time, the Croatian Chamber of Dental Medicine published a statement on its website related to working in dental institutions for the treatment of healthcare professionals in case of suspected COVID-19. In mid-March, the National Crisis Headquarters decided to limit social gatherings, service activities, and sports and cultural events. The provision of health services was also limited. The state introduced financial assistance for business entities during the lockdown period.

After two months, preventive public health measures were eased, and the public and private health system were enabled to work at full capacity, as a result of which dental medicine offices and dental laboratories started working, with mandatory compliance with special epidemiological measures. On 8 May 2020, guidelines and protocols for the opening and operation of dental practices and dental laboratories were published on the website of the Croatian Chamber of Dental Medicine. The main advice for prevention was frequent hand washing; disinfection of frequently touched surfaces; covering the mouth and nose with an elbow or tissue when coughing and sneezing; and avoiding touching the eyes, nose, and mouth with hands. It was also recommended to wear FFP2 masks, ventilate closed spaces, maintain physical distance, avoid close contact with other people, and avoid crowds and large gatherings [7].

A new infectious disease has introduced a lot of unknowns and uncertainties to healthcare workers. In their daily work, dentists have an increased risk of infection due to contact with patients, as it is impossible to maintain the recommended distance, and the work itself includes contact with patient secretions [8]. Additionally, rotating and ultrasonic instruments used in dental practices generate aerosols and promote the spread of viruses [9]. Therefore, additional prevention measures are necessary, including personal protection of staff; disinfection; sterilization of all surfaces, devices, and instruments in the office; decontamination procedures; ventilation of the office after each patient; and prior assessment of the patient [10]. As for the patient himself, it is very important to take his detailed history, carry out triage, performed adequate assessment and admission of the patient, measure his body temperature, and attempt to reduce his stay in the office as much as possible [11]. During triage, each patient is asked whether he has had a fever or respiratory symptoms in the past 14 days, whether he has traveled abroad in the past 14 days, or had been in contact with a febrile person who had breathing difficulties and a cough, and whether the patient participated in a large gathering [11].

The aim of this research was to analyze how the COVID-19 pandemic affected the activity of dental medicine in the Republic of Croatia in 2020. The hypotheses were that:The COVID-19 pandemic had a larger impact (financial impact and change in the number of visits) on dentists with a higher percentage of income from dental tourism; and tharDentists’ fear is associated with previous COVID-19 infection, employment status, and public health insurance contracts.

Moreover, in this study, we aimed to test possible predictive factors (years of work experience, contract with health insurance, percentage of profit from health tourism, previous self-isolation and infection, symptomatology, use of protective equipment, patient triage, drop in visits to the practice, change in the provision of services, the decision to get vaccinated, and wearing a disposable coat/apron) for dentists’ fear.

## 2. Materials and Methods

This cross-sectional research included a survey that was carried out via Google Forms. The survey was created by the authors and consisted of 27 questions. It included questions about years of work experience, type of practice (private or public), location of the dental office, percentage of profit from health tourism, previous self-isolation and infection, source of infection, symptomatology, use of protective equipment, patient triage, number of visits to the practice, provision of services, attitudes about vaccination, and five elements of fear (fear that there are not enough protective measures for the dental profession, fear of falling ill due to the nature of the job, fear for one’s own life, fear of loved ones falling ill, and fear for financial existence) (Appendix A).

The link to the online survey was sent by e-mail to all doctors of dental medicine who are members of the Croatian Chamber of Dental Medicine. Sample size calculation and sampling were not applied. Out of 5066 doctors of dental medicine, 136 completed the survey (2.68%).

The survey period was 15 days, from 27 January 2021 to 10 February 2021.

The study’s primary outcome was to assess the changes in the dental profession during the pandemic, more precisely the dentists’ fear.

Data were analyzed using IBM SPSS 22 (IBM Corp., Armonk, NY, USA), applying the following statistical tests: Shapiro–Wilk test, Chi-square test, Fisher’s exact test, *t*-test, Mann–Whitney test, analysis of variance with Student–Newman–Keuls post hoc test, Kruskal–Wallis test with Mann–Whitney post hoc test and Bonferroni correction for multiple comparisons, Spearman’s correlation, exploratory factor analysis, and linear regression. Correlations were interpreted at r > 0.250 with *p* < 0.05.

The research was approved by the Ethical Committee of the Faculty of Dental Medicine, Rijeka (class: 035-01/21-01/02, urbr: 2170-57-006-21-1).

## 3. Results

More participants were from the coastal parts of Croatia (65%) than from the continental part. The majority of participants (74%) were employees in dental practices, and 24% of participants were the owners of practices. They worked mainly in offices that have a contract with public health insurance (65%). A proportion of 56% of the respondents had income from health tourism, and the income mostly accounted for up to 10% of the total profit of the practice. Younger dentists dominated, and 69% of participants had fewer than 20 years of experience.

The transmission of the coronavirus from the dentist to the patient did not occur in any case. The majority of dentists (65%) planned to get vaccinated against the coronavirus.

As for the protective equipment that dentists used daily in their work, they all used gloves, 99% used a protective mask (mainly surgical, 78%), 90% used a face shield, 57% used a surgical cap, 23.9% used a disposable coat or apron, and 38.4% used a reusable coat. In terms of protective measures, 99% of dentists washed their hands several times every day at the workplace, 96% ventilated the rooms used to provide dental services, and 85% used enhanced disinfection measures. A proportion of 75% of dentists triaged patients using a questionnaire, and 79% measured patients’ body temperature.

A proportion of 49% of dentists noticed a decrease in office visits. Minor changes in clinical work were observed by 43% and major changes by 9% of participants. There was less fear that due to the nature of the work, they would get sick with COVID-19 (median score of 1 on a scale from 0 = no fear all to 10 = extreme fear), as well as the fear that due to the nature of the work, they would not have sufficient available protection measures against COVID-19 and fear for one’s own life due to the nature of the work due to the epidemic (median score of 3). There was a moderate fear of financial existence and that due to the nature of the job, their loved ones (family members and friends) would get sick (median score of 5 on a scale of 0–10).

Doctors of dental medicine who noticed a decrease in office visits were more often those without a contract with public health insurance than those who had a contract (70% vs. 37%, respectively; *p* < 0.001), and they also had a higher share of profit from health tourism (32% vs. 14%, respectively; *p* < 0.001). Fear of the COVID-19 disease was not related to previous infection or a certain measure of self-isolation.

There was no statistically significant difference in the fear between the owners and employees of dental offices. Among dentists who do not have a contract with public insurance, fear of financial existence was more pronounced than among those who have a contract (*p* = 0.002).

Fear of financial existence due to the COVID-19 epidemic is positively linearly correlated with the percentage of profit from health tourism, but the correlation is weak (r = 0.299; *p* < 0.001; Figure 1). As the percentage of profit from health tourism increases, so does fear. However, there was no association between other types of fear and the percentage of profit from health tourism. Years of work experience were also unrelated to fear.

Those who had a greater fear that they or their loved ones would get sick due to the nature of their work planned to get vaccinated (*p* ≤ 0.007; Figure 2). The decrease in office visits was not significantly related to the decision to vaccinate.

Exploratory factor analysis revealed that five elements of fear form one dimension explaining 68% of the variance. Therefore, the sum of answers to those five questions was used in multiple linear regression to explore predictors of fear of COVID-19 infection. Only the decision to get vaccinated and wearing disposable coat/apron were related to fear when all other parameters were controlled for (years of work experience, contract with health insurance, percentage of profit from health tourism, previous self-isolation and infection, symptomatology, use of protective equipment, patient triage, drop in visits to the practice, and change in provision of services; R = 0.44; *p* = 0.037).

## 4. Discussion

This research shows that the COVID-19 pandemic did not greatly affect the activity of dental medicine in the Republic of Croatia. Although dentists are extremely exposed to the risk of infection in the performance of their work—according to research, the profession is the most exposed to the risk of infection (more so than doctors of general medicine and medical technicians)—they are used to protective measures and were not significantly concerned about safety [12].

Infection and isolation among dentists in Croatia did not occur in great numbers and included mild symptoms. The source of infection was more often outside the dental office, and there was no transmission of the coronavirus to the patient. In a study conducted in Iran, only 1% of examined dentists tested positive for the coronavirus, whereas 7% noticed symptoms of COVID-19 [13]. Additionally, only 3% of the examined dental assistants had symptoms, but none of them had a positive result in a coronavirus test [13]. A very low incidence of infection with the SAR-CoV-2 virus among doctors of dental medicine (5.10 per 100,000) was also reported in a study conducted on doctors in the public health system in Canada [14].

The COVID-19 pandemic, as a newer form of occupational disease in dental medicine, has a certain impact on the performance of dental work. Dentists use gloves, masks, and face shields in their work as a standard, and during the pandemic, they additionally used aprons, coats, and caps. Among the additional measures, regular hygienic hand washing, airing of rooms, and increased measures of disinfection of premises and dental equipment were carried out. In most offices, triage and measurement of the body temperature of patients were carried out. In Iran, most dentists followed the latest prevention and protection measures and implemented them in their work [13], as well as in Saudi Arabia, where dentists believe they are prepared to work during the pandemic [15]. In Italy, increased use of face shields, surgical caps, disposable gowns, preoperative mouthwashes, and rubber dams was recorded in comparison to the prepandemic period [16]. On the other hand, in Poland, the dental sector believes that it is not ready to work in a pandemic, and the majority of practitioners decided to close their offices. The reason for this is the insufficient coordination of the health system and the lack of protective equipment, which resulted in fear, insecurity, and anxiety among doctors of dental medicine [17].

Usual protective measures are not sufficient to protect dental workers from infection. This particularly applies to dental procedures that generate aerosol, and during the pandemic, preference has been given to procedures that do not generate aerosol, for example, manual excavation of caries or hand root canal instrumentation instead of rotatory and reciprocal techniques [18]. In Italy, aerosolization (i.e., ultrasound) was drastically reduced during the pandemic [16]. Although several reports have suggested various measures, a fully effective system of protective measures for dental workers is clearly lacking [19].

In the USA, there is great concern among healthcare workers when performing jobs that predispose them to viral infection, considering that healthcare costs are higher and that protective equipment is lacking [20]. The change in healthcare and even in dental medicine happened almost overnight, and the use of protective equipment became necessary in order for dental workers to protect themselves and their patients [21].

The drop in visits was greater in practices that do not have a contract with public health insurance and that are engaged in dental tourism. The first hypothesis is accepted, i.e., that the COVID-19 pandemic had a greater impact on dentists that are more engaged in dental tourism. According to the data of the State Institute for Public Health, the decline in dental visits was about 10% [22,23,24,25,26]. A survey conducted among dentists in Iran showed that 95% of participants limited working hours and patient visits to emergencies only [13]. Moreover, most dentists postponed any non-urgent procedures during the pandemic. Such practice is very positive, considering that reducing the number of patients coming to the office is important in order to avoid possible cross-infections between patients, and is easier to maintain the recommended social distance [19]. A similar practice for postponing dental visits was implemented in Italy [16]. Furthermore, with such measures, more time is available for the disinfection of the office.

Dentists in Croatia were more afraid that their loved ones would get sick than themselves. Research by Consolo et al. showed greater concern for the possibility of infection during work (≈85%) and a negative impact of the COVID-19 pandemic on the psychological state of dentists in the form of feelings of fear, worry, and anxiety. A higher level of negative impact is expected, considering that the research was conducted on a sample of dentists from one of the Italian regions that were most affected by the pandemic [27]. In a survey conducted among dentists, dental assistants, and hygienists in Norway, more than 70% expressed fear that they would get infected, that they would infect others, or that their family members would get infected [28]. Nevertheless, a minimal number of participants believed that the future of their work was threatened due to the COVID-19 pandemic [28]. According to research conducted in approximately 30 countries around the world, 87% of participants feared contracting COVID-19 from a patient or colleague [29]. As many as 90% of participants were anxious, and more than 72% of participants felt uncomfortable when talking to patients in the immediate vicinity. A proportion of 92% of participants were afraid of transmitting an infection from the dental office to their family, and 77% felt afraid of being quarantined. A significant number of dentists considered closing their dental practice until the number of COVID-19 cases began to decrease [29]. Similar results were published in a study conducted in Texas, USA, according to which most dentists were concerned about their health and safety (77%) and the future of their practices (90%). Likewise, a large proportion (≈60%) believed that work became extremely demanding during the pandemic, but the vast majority (96% to 98%) felt confident in continuing to perform their work and implementing protective measures during the pandemic [30].

Dentists’ fear was not associated with employment status and previous positive test results for SARS-CoV-2 or previous self-isolation (alternative hypothesis is rejected), but was significantly associated with having a contract with public health insurance (hypothesis accepted). The fear of the financial existence of dentists in Croatia was moderate and greater than the fear of getting sick. This fear was not related to the ownership of a practice, but it was related to secure sources of income. Those without contracts with public insurance and with higher profits from dental tourism had a more pronounced fear. In a survey conducted among German dental workers, a large impact of the COVID-19 pandemic on monetary profits was observed [19]. In particular, a large number of dental offices in Germany are facing the problem of reduced profits or minimal earnings compared to previous years.

Research conducted in Australia showed the positive effects of the COVID-19 pandemic on the dental profession, such as determining and adhering to infection control and prevention measures, reducing workload and providing support, and achieving cooperation and teamwork within the profession and in general health care [31].

Despite minor or moderate fear among dentists in Croatia, participants wanted and planned to be vaccinated against the coronavirus. A study conducted in Poland showed a significantly lower percentage (44.5%) [32]. The reason for greater motivation to get vaccinated is primarily attributed to knowledge about the benefits of vaccines propagated by the health profession to which they belong. It is to be assumed that a greater number of doctors would also decide to get vaccinated, as this research was conducted at a time when the first vaccines had just begun to arrive in Croatia.

## 5. Conclusions

It seems that the COVID-19 pandemic had a minimal impact on the dental profession in Croatia, as protective and infection control measures were standard in dental medicine on a daily basis even before the pandemic. The fear of financial existence due to the epidemic of COVID-19 is connected with the absence of contracts with public health insurance and health tourism. Vaccination is motivated by the fear of infecting oneself and loved ones.

The small sample size and lack of validation process for the questionnaire represent limitations of this study. Because this study is the first to be published about the COVID-19 pandemic’s impact on the dental profession in Croatia and it included the whole national database of doctors of dental medicine, it provides an initial estimate and foundation for further research, which is its greatest strength. We suggest that further studies be conducted with a similar study design to obtain results with a larger sample of participants in order to overcome the limitations of a small number of participants in this study.

## Figures and Tables

**Figure 1 dentistry-11-00009-f001:**
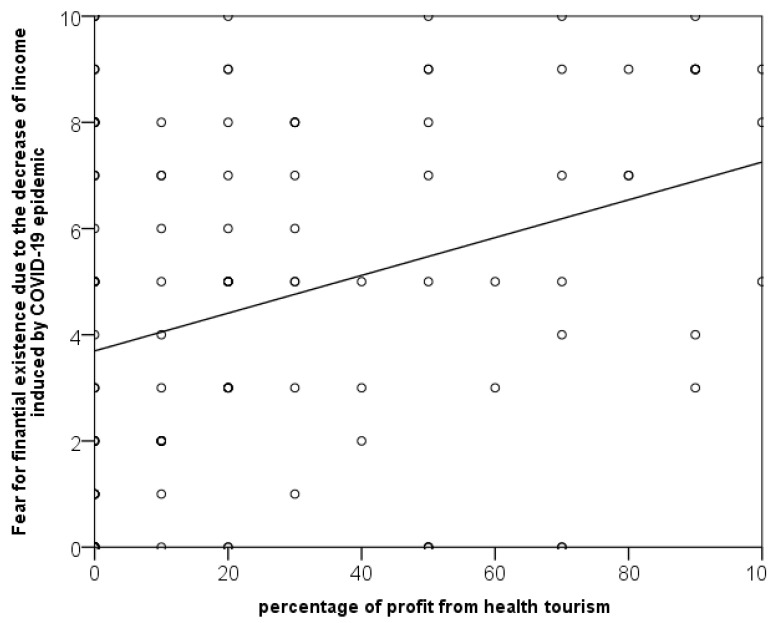
Scatter plot presenting the association of profit from health tourism with fear of financial existence due to the COVID-19 pandemic, with the line demonstrating a linear trend.

**Figure 2 dentistry-11-00009-f002:**
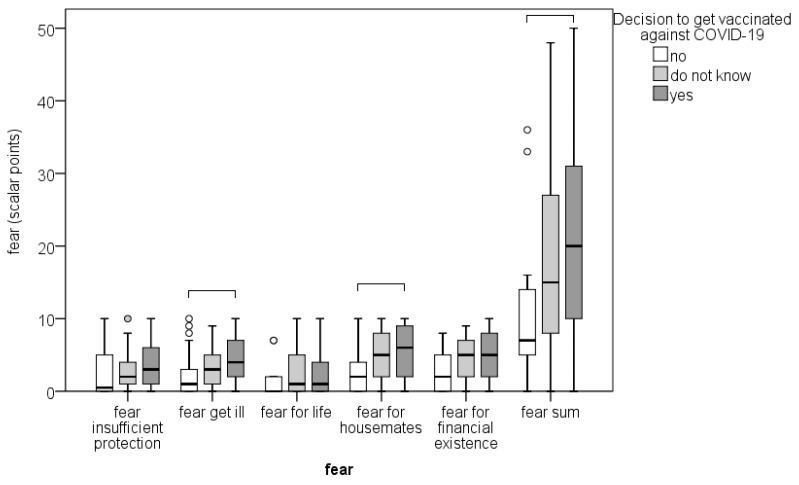
Association of dentists’ fear (scale 0 = no fear to 10 = extreme fear) with the decision to get vaccinated. Box plot represents the median and interquartile range, whiskers represent min and max values, and circles represent outliers. Parentheses connect categories that differ significantly according to Kruskal–Wallis test with Mann–Whitney post hoc test and Bonferroni correction for multiple comparisons.

## Data Availability

Not available.

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
