# Peer review of "Pandemic Financial Stress in Dental Medicine in Croatia"

_dentistry, 2022, doi:10.3390/dj11010009_

Round 1

Reviewer 1 Report

The article is very interesting, the research is nicely performed, and the argument could be very interesting for the readers.

However, before publication, I would suggest some improvements.

1) Line 80:

The authors declared a (null?) hypothesis.

It shall be accepted or rejected in the discussion.

The authors could also consider splitting the hypothesis in more than one.

2) Line 88:

The authors reported that they sent messages to members of the Croatian Chamber of Dental Medicine.

Please specify the numbers so that the percentage of dental professionals responding to the survey can be added to the results (136/numbers of messages sent)

3) Line 95:

Please specify the start and the end of the survey period (how many days in total)

4) Line 110:

The authors wrote:

“In none of the participants, the transmission of the coronavirus to the patient occured.”

Please explain better this sentence.

5) Lines 175-198:

The authors could consider also the findings of another study that has several similarities.

It is another survey on dental professional, google forms, and analyze Convid-19 pandemic situation, consequences on stress, protective measures, etc.

Paolone G, Mazzitelli C, Formiga S, Kaitsas F, Breschi L, Mazzoni A, Tete G, Polizzi E, Gherlone E, Cantatore G. One-year impact of COVID-19 pandemic on Italian dental professionals: a cross-sectional survey. Minerva Dent Oral Sci. 2022 Aug;71(4):212-222. doi: 10.23736/S2724-6329.21.04632-5. Epub 2021 Dec 1. PMID: 34851068.

Taken from the abstract:

A total of 614 between dentists and dental hygienists completed the questionnaire. Compared to the prepandemic period, the use of PPE such as face shields, surgical caps and disposable gowns were implemented after the COVID-19 outbreak. Almost the whole interviewed (99.9%) received the COVID-19 vaccine. An increased use of preoperatory mouthwashes and rubber dam was referred during the pandemic, while aerosolization (i.e., ultrasound) was drastically reduced. A certain number of respondents (30% dentists, 27% hygienists) suffered from work-related stresses during the pandemic until they desired to change jobs. E-learning was considered beneficial for the 70% of participants.

Therfore, in addition to Iran, Poland and Canada, please add a reference to Italy (country that suffered high consequences)

6) Line 203: 

postponed treatments: again lots of similiarities to the previous article

7)Lines 242-248:

The autohrs could suggest further studies to be conducted such as repeating a similar survey in the near future.

Author Response

1) Aim of the study and hypothesis are more thoroughly explained. The alternative hypothesis is split into two hypotheses. Hypotheses are now accepted and rejected in the discussion.

2) Number of all the members of the Croatian Chamber of Dental Medicine and the percentage of doctors that took part in the study are now shown.

3) The total number of days of the survey period and exact dates are now added.

4) Sentence “In none of the participants, the transmission of the coronavirus to the patient occured.” is rewritten.

5) and 6)

Study by Paolone et al. is inluded in the Discussion part, protective measures, decrease in aerosolization and postponed treatments are compared with the results from this study.

7) Suggestions for further studies are added in the conclusion.

Reviewer 2 Report

It is not specified how the sample size was calculated.

No sampling technique

The study lacks clinical relevance, it is too descriptive. No multivariate analysis has been considered to determine associated factors.

Nothing is mentioned about the statistical validation of the questionnaire.

Apparently there is a bias by not considering the validation process of the instrument.

Author Response

Sample size calculation was not performed - questionnaires were sent to all dentists in Croatia. No sampling technique was used.

New analyses are now added (multiple regression).

Exploratory factor analysis is now added for the dimensions of fear. Other questions in the survey were simple and direct and were used mainly as predictors.

Questionnaire was not validated, and it is one of the limitations of the study.

Reviewer 3 Report

Thank you for allowing me to review this Article, the purpose of which was to investigate how the COVID-19 pandemic affected the activity of dental medicine in the Republic of Croatia in 2020. The topic is interesting and important; however, there are areas for improvement in the implementation that must be corrected before publication. 

1. In the title or abstract of the Article, the authors did not indicate the study's design.

2. The results in the abstract are descriptive; please state the results in numbers.

3. In conclusion, the word small effect may not be the best choice; please find a synonym.

4. In the work methodology, state the period in which study was carried out.

5. How was the sample size calculated? How many members are there in the Chamber of Dental Medicine in Croatia?

6. What are the inclusion and exclusion criteria?

7. Who designed the questionnaire, how many questions, and how long did it last? Who validated the questionnaire?

The methodology section needs to be more suitable, and it should be more detailed!

8. Explain the study's primary outcome more clearly in the methodology?

9. Please describe any efforts to address potential sources of bias.

10. What statistical method was used to determine the data distribution? How was the data displayed? Which data were processed by which statistical test?

11. Please show the socio-demographic data of the respondents in a table.

12. In each table at the bottom, describe by which method the results were analyzed.

13. Couldn't the results of the tables be shown more simply by regression or correlation or by some other comparable method, but all the tables are similar and quite dull?

14. In the introduction, the authors stated: "The hypothesis was that the pandemic increased protection measures, disinfection, and costs, reduced the income of surgeries, increased concern about one's illness and the illness of loved ones, and loss of income. It is expected that the fear is greater in those with more work experience, those who do not have a contract with health insurance, and those with a higher percentage of income from health tourism." - however, almost none of this is shown in the results. Let the authors look at what they researched and presented in the results.

15. What are the limitations of the study? And what is its strength, if it has any at all?

16. References are not written following the instructions of the Journal.

The research was carried out very simply on a small number of respondents. I asked the authors to attach the questionnaire so that it would be possible to check what they asked the respondents and to follow the course of the Article more easily.

Author Response

  1. The study's design is now indicated in the abstract.
  2. The results in the abstract are now stated in numbers.
  3. In conclusion, the word small effect is changed with low impact.
  4. The period in which the study was carried out is stated in the methodology.
  5. Sample size calculation was not performed - questionnaires were sent to all dentists in Croatia (5066). The number of all the members of the Croatian Chamber of Dental Medicine and the percentage of doctors that took part in the study are now shown.
  6. Inclusion criteria: all the doctors of dental medicine who are working as DMD in Croatia (they must be registered in Croatian Chamber of Dental Medicine) and who were willing to take part in the study.
  7. The questionnaire was designed by authors (Renata Grzic and Stjepan Spalj). It consisted of 24 questions, it took around 10 minutes to fill in the questionnaire and it was not validated.

The methodology section is now explained in more detail.

  1. Primary outcome of the study is now stated in the methodology section,
  2. Possible sources of bias in this study were selection bias and reporting bias. Efforts to address the reporting bias were made and all the results including statistically significant and non statistically significant are shown in the Result section. Selection bias could not be addressed more because all the doctors in Croatia who were willing to take part in the study were included but further studies are recommended in the conclusion.
  3. Visual inspection of histogram and Shapiro-Wilk test were used to check to test distribution of data. When data followed normal or at least symmetrical distribution parametric tests were used, otherwise non-parametric test were applied. Statistical methods are now supplemented.
  4. Age and gender were not registrated, just years of experience which is reported in text.
  5. Tables are now omitted and statistical methods are added in figure legends.
  6. Tables are omitted and figures are added.
  7. Aim of the study and hypothesis are more thoroughly explained, and are in line with the results.
  8. Limitations and strengths of the study are shown in the final paragraph.
  9. References are now rewritten, following the instructions of the Journal.

A small number of participants is listed as the limitation of the study. The questionnaire is attached.

Round 2

Reviewer 2 Report

Now the study is well organized

Author Response

Thank you for your comments and advices!